# Differences in Morphokinetic Parameters and Incidence of Multinucleations in Human Embryos of Genetically Normal, Abnormal and Euploid Embryos Leading to Clinical Pregnancy

**DOI:** 10.3390/jcm10215173

**Published:** 2021-11-05

**Authors:** Katerina Tvrdonova, Silvie Belaskova, Tatana Rumpikova, Alice Malenovska, David Rumpik, Alena Myslivcova Fucikova, Frantisek Malir

**Affiliations:** 1Department of Biology, Faculty of Sciences, University of Hradec Kralove, 500 03 Hradec Kralove, Czech Republic; alena.fucikova@uhk.cz (A.M.F.); frantisek.malir@uhk.cz (F.M.); 2Clinic of Reproductive Medicine and Gynecology Zlin, U Lomu 638, 760 01 Zlin, Czech Republic; tatana.rumpikova@ivfzlin.cz (T.R.); david.rumpik@ivfzlin.cz (D.R.); 3Institute of Mathematics and Statistics, Faculty of Science, Masaryk University, 611 37 Brno, Czech Republic; silvie.belaskova@fnusa.cz; 4Association of Reproductive Embryology, Seifertova 801/64, 633 00 Brno, Czech Republic; malenovska@hotmail.com

**Keywords:** time-lapse monitoring, morphokinetics, multinucleation, embryo development, preimplantation genetic testing

## Abstract

The selection of the best embryo for embryo transfer (ET) is one of the most important steps in IVF (in vitro fertilisation) treatment. Preimplantation genetic testing (PGT) is an invasive method that can greatly facilitate the decision about the best embryo. An alternative way to select the embryo with the greatest implantation potential is by cultivation in a time-lapse system, which can offer several predictive factors. Non-invasive time-lapse monitoring can be used to select quality embryos with high implantation potential under stable culture conditions. The embryo for ET can then be selected based on the determined morphokinetic parameters and morphological features, which according to our results predict a higher implantation potential. This study included a total of 1027 morphologically high-quality embryos (552 normal and 475 abnormal PGT-tested embryos) from 296 patients (01/2016–06/2021). All embryos were cultivated in a time-lapse incubator and PGT biopsy of trophectoderm cells on D5 or D6 was performed. Significant differences were found in the morphological parameters cc2, t5 and tSB and the occurrence of multinucleations in the stage of two-cell and four-cell embryos between the group of genetically normal embryos and abnormal embryos. At the same time, significant differences in the morphological parameters cc2, t5 and tSB and the occurrence of multinucleations in the two-cell and four-cell embryo stage were found between the group of genetically normal embryos that led to clinical pregnancy after ET and the group of abnormal embryos. From the morphokinetic data found in the PGT-A group of normal embryos leading to clinical pregnancy, time intervals were determined based on statistical analysis, which should predict embryos with high implantation potential. Out of a total of 218 euploid embryos, which were transferred into the uterus after thawing (single frozen embryo transfer), clinical pregnancy was confirmed in 119 embryos (54.6%). Our results show that according to the morphokinetic parameters (cc2, t5, tSB) and the occurrence of multinucleations during the first two cell divisions, the best euploid embryo for ET can be selected with high probability.

## 1. Introduction

The goal of assisted reproduction is for the patient to give birth to a healthy child. Selecting the best embryo for embryo transfer is a long-standing effort of IVF (in vitro fertilisation) specialists. Today, in most cases, only the best embryo is transferred (elective single embryo transfer, eSET) to avoid unwanted multiple pregnancies [1,2].

Embryo selection can be performed using invasive or non-invasive technologies. Non-invasive methods include embryo morphology assessment, time-lapse monitoring, metabolomics and proteomics. Invasive techniques include embryo biopsy for genetic testing [3].

One of the important indicators of the quality of an embryo is its morphological parameters. The evaluation of preimplantation embryo morphology is subjective and can be misleading [4]. Thanks to time-lapse incubators, it is possible to evaluate these parameters without exposing developing embryos to conditions outside the incubators, where they have an ideal environment for their development [5,6]. Thanks to this technology, it is possible to evaluate embryos continuously because images are always taken at several-minute intervals [2,7]. Continuous culture time-lapse systems have been identified as safe for embryo culture [8] and potentially shape a cultivation environment leading to increased blastocyst formation [8,9] and higher rates of implantation and clinical pregnancy [7,8,10]. Thanks to the cultivation of embryos in time-lapse systems, it is possible to obtain very valuable information about the morphokinetics of the embryo and other developmental markers, such as fragmentation and multinucleations [11], which cannot be easily registered in traditional assessment under a microscope at 24-h intervals.

Because aneuploidies are the most common genetic abnormality in humans and more than 50% of IVF embryos are aneuploid [12,13], preimplantation genetic testing for aneuploidies (PGT-A) is now a very well-established technique. In the last 15 years, new technologies based on whole genome amplification (CGH—Comparative Genome Hybridization, NGS—Next Generation Sequencing) have been implemented, thus expanding the possibilities of the genetic testing of embryo aneuploidies, especially increasing the ability to recognize embryonic mosaicism [14,15]. According to the recommendation of PGDIS (Preimplantation Genetic Diagnosis International Society, 2019), embryos up to a maximum of 20% of the mosaic are considered genetically normal.

Thanks to PGT-A, which can determine the ploidy of embryos, it is possible to increase the success of IVF treatment [16,17]. With the implementation of trophectoderm blastocyst biopsy on the fifth or sixth day of the culture (not only one or two blastomers on the third day of embryo development) in IVF treatment cycles with PGT-A and using whole embryo genome amplification, a higher pregnancy rate was observed, especially in cycles with vitrified biopsied embryo transfer compared to cycles with fresh embryo transfer in the morning of the sixth day of the culture (with a biopsy in the morning of the fifth day of the culture) [18].

The aim of this work is to determine whether there is a relationship between genetic results after PGT-A examination (normal-euploid embryo or abnormal-aneuploid embryo) in 5- to 6-day-old human embryos cultured in a time-lapse system and morphokinetic parameters (cc2—durations of the second cell cycle, s2—duration of the 3rd mitosis, t5—time of reaching the five-cell embryo stage, tSB—time of reaching initiation of blastulation) and the occurrence of multinucleations in the two-cell embryo and four-cell embryo. In addition, an attempt is being made to determine the relationship between morphokinetic parameters and the occurrence of multinucleations in genetically normal embryos that led to clinical pregnancy after subsequent frozen embryo transfer of the examined embryo and genetically abnormal embryos. Based on the obtained results, the aim of the work is also to design time intervals for morphokinetic parameters that could be useful for the selection of embryos with high implantation potential for embryo transfer in patients who do not want or cannot undergo PGT-A.

## 2. Materials and Methods

This retrospective cohort study is based on data from the database of the Clinic of Reproductive Medicine and Gynecology Zlín. The study included patients who underwent treatment at our IVF clinic with pre-implantation genetic testing of embryos (PGT-A) and culture of embryos in a time-lapse system in the period 01/2016 to 06/2021. There were a total of 296 patients, mean age 32.7 ± 4.5 years. The indication for PGT-A was usually older maternal age, miscarriages, or failure of implantation in previous IVF cycles.

Ovarian stimulation and oocyte collection were performed according to standard treatment protocols. In most cases, ovarian stimulation was performed using the GnRH antagonist protocol, Decapeptyl or Dipherelin was applied to induce ovulation, and oocyte collection was under ultrasound control 36 h after trigger application [19,20].

All oocytes were denuded and ICSI (Intracytoplasmic Sperm Injection) was performed on mature oocytes (MII) [21] 4 h after oocyte collection. All embryos were cultured individually in an EmbryoSlide (Vitrolife, Göteborg, Sweden) culture dish with one-step G-TL medium (medium for time-lapse culture) (Vitrolife, and the dish was overlaid with 1.5 mL of mineral oil (Ovoil, Vitrolife). Embryos were cultured in a time-lapse system (EmbryoScope, Vitrolife) at 37 °C, 6% CO_2_, 5% O_2_ to blastocyst or expanded blastocyst stage (D5, D6). Laser-assisted hatching (OCTAX, Microscience GmbH; Bruckberg, Germany) was performed on D3 to open the zona pellucida so that the trophectodermal cells could herniate from the zona pellucida [22].

The study included a total of 1027 morphologically high-quality embryos (552 euploid and 475 aneuploid PGT-A-tested embryos) from 296 patients. The biopsy of the trophectoderm was performed on all quality embryos on D5 or D6 (4AA, 4AB, 4BA, 5AA, 5AB, 5BA, 6AA, 6AB, 6BA) according to Gardner and Schollcraft [1] once the trophectoderm cells sufficiently herniated from the zona pellucida (minimum 10 cells) using a laser (OCTAX). Biopsied trophectoderm cells (5–7) were washed according to standard protocol in PVP buffer (1% polyvinylpyrrolidone), placed in a PCR microtube with 2 µL PBS buffer, frozen at −20 °C and transferred to an accredited molecular genetic laboratory, where SurePlex DNA isolation and whole genome amplification were performed (SurePlex DNA Amplification system, Illumina, San Diego, CA, USA). The obtained DNA sample with a final concentration of about 50 ng/µL was examined by arrayCGH (DNA labeling, on-chip hybridization—24 sure+, BlueGnome, Illumina, analysis using BlueFuse Multi software) or the chromosomal profile of the samples was determined by NGS (VeriSeq PGS Kit, Illumina). The principle of this method is the preparation of a sequencing library, where individual samples are marked with specific indices, which is then amplified and sequenced by SBS (MiSeq, Illumina). Biopsied embryos were vitrified 1 h after biopsy (Rapid VitBlast, Vitrolife) [23].

On all embryos, PGT-A was performed, the image from the EmbryoScope (image every 10 min, 7 focal planes) was analyzed and all known morphokinetic and morphological data were read using EmbryoViewer software. The parameters used for the study were: duration of the second cell cycle (cc2), duration of the third mitosis—the time between the division of the 3-cell embryo into a 4-cell embryo (s2), time to the division of the embryo into a 5-cell embryo (t5), time until the onset of blastulation (tSB). The time of cell division is the moment when a complete septum is formed from the cytoplasmic membrane and the two new blastomeres are completely separated. The blastulation initiation time (tSB) is when the first blastocoel cavity forms between embryonic cells. The incidence of multinucleations in the two-cell and four-cell stages of the blastomere was further evaluated, with each embryo with multinucleations in at least one blastomere being considered an embryo with multinucleations. The annotation was performed all the time by the same experienced embryologist. Following the menstrual cycle, the patient was prepared for FET (frozen embryo transfer) of one euploid embryo using an HRT (hormone replacement therapy) cycle (from the 1st day of the menstrual cycle estrogens (Estrofem) are used, from the 3rd day the transdermal estrogen spray (Lenzetto) is started, on the 13th day of the cycle the endometrial thickness is checked by ultrasound (if it is more than 7 mm the patient starts taking gestagens) and on the fifth or the sixth day of use FET is scheduled according to the age of the embryo) or a native cycle (natural menstrual cycle with ultrasound control of endometrial thickness on day 13 of the cycle; if the endometrium is at least 7 mm and the dominant follicle is larger than 15 mm, the patient starts taking gestagen from the following day, and FET is scheduled for the 5th or 6th day of use according to the age of the embryo).

Based on the result of the PGT-A examination, 2 groups of embryos were created: euploid (normal) embryos (*n* = 552) and aneuploid (abnormal) embryos (*n* = 475). After performing KET, the result of the pregnancy test was recorded, and if it was positive, the patient was monitored until clinical pregnancy was determined by ultrasound (fetal heartbeat)—these embryos that gave rise to clinical pregnancy were included in the third group—euploid embryos with proven clinical pregnancy (*n* = 119).

Continuous data from the time-lapse analysis were interpreted as mean ± SD (standard deviation) with respect to data distribution. Otherwise, they were interpreted as median and inter-quartile ranges (p25—lower quartile, p75—upper quartile). Two sample *t*-tests were used for testing the hypothesis about the means of time lapses of cell cycles. In the case of skewed data, the alternative non-parametric approach was used—Mann–Whitney *U*-test. Logistic regression was used to calculate the OR (odds ratio) along with a 95% CI (confidence interval). The Shapiro–Wilk test was used to test the normality of the data. Bootstrap 95% CI was calculated for the comparison of time intervals of euploid and aneuploid embryos. A standard significant level of significance was chosen to reject the null hypothesis of 0.05. All tests were two-sided. SAS software 9.4 and R software were used for all analyses.

## 3. Results

The results of the study show that PGT-A normal embryos have significantly different durations of the second cell cycle (cc2) (*p* = 0.007), significantly different times of reaching the five-cell embryo stage (t5) (*p* = 0.002) and a significantly different time to blastulation initiation (tSB) (*p* < 0.0001) compared to the PGT-A group of abnormal embryos (aneuploid). For parameter s2, the time between division from a three-cell to a four-cell embryo (time between second and third mitosis) was not significantly different in these two groups, but a trend was observed—longer duration between division from a three-cell to a four-cell embryo in the group of aneuploid embryos (Table 1, Figure 1, Figure 2 and Figure 3).

In the groups of euploid embryos with proven clinical pregnancy and aneuploid embryos, there are significantly different durations of the second cell cycle (cc2) (*p* = 0.006), significantly different times of reaching the five-cell embryo stage (t5) (*p* = 0.004) and significantly different times of reaching the initiation of blastulation (tSB) (*p* < 0.0001). For parameter s2, the length between the division from a three-cell to a four-cell embryo (time between second and third mitosis) was also significantly different in the two groups (*p* = 0.021). (Table 2, Figure 1, Figure 2 and Figure 3).

The results of the comparison of the groups of euploid embryos and aneuploid embryos in the incidence of multinucleations show that in the group of euploid embryos, multinucleations in the two-cell stage occur 2× (OR = 2.18, *p* < 0.0001) less frequently than in the group of aneuploid embryos. Compared to the incidence of multinucleations in the four-cell stage, multinucleations in the group of euploid embryos occur 6× (OR = 6.27, *p* < 0.0001) less frequently compared to the group of aneuploid embryos (Table 3).

In the group of euploid embryos with a proven clinical pregnancy, multinucleation in the two-cell stage embryo occurs 2× less frequently (OR = 2.013, *p* < 0.0001) than in the group of aneuploid embryos. Regarding multinucleation in the group of euploid embryos with proven clinical pregnancy, multinucleations in the four-cell embryo occur 7× less frequently (OR = 7.016, *p* <0.0001) than in the group of aneuploid embryos (Table 3).

From the data obtained in our study, predictive time intervals for the selection of euploid embryos with high implantation potential were determined: cc2: 11.3–11.7 h, t5: 48.1–49.5 hpi, tSB: 92.9–94.6 hpi (Figure 4, Figure 5 and Figure 6).

Finally, the morphokinetic parameters in the group of euploid embryos with proven clinical pregnancy and the group of euploid embryos without implantation after ET were compared. Significant differences were found in the morphokinetic parameters t5 (*p* = 0.037) and tSB (*p* = 0.036).

## 4. Discussion

In this study, embryos are evaluated using the PGT-A method and time-lapse monitoring with different morphokinetic and morphological results in the PGT-A group of abnormal embryos and normal embryos, as well as in the group of normal embryos that led to clinical pregnancy.

The incidence of aneuploidies in human oocytes and embryos, which affect more than 50% of embryos and whose incidence increases with maternal age, is a major reason for implantation failure and abortions in IVF cycles [24]. Embryos for embryo transfer tend to be selected on the basis of their morphology, but it has been shown that even aneuploid embryos are able to achieve a high morphological score [25].

At present, the evaluation of embryos using time-lapse monitoring is increasingly developing. Quite a number of studies have focused on determining the predictive factors obtained from cultivation in a time-lapse culture system to label a euploid embryo, but their results have been different [26].

Davis et al. [27] found a delay in the first and second cell division and a prolongation of the time between the two-cell stage and four-cell stage in aneuploid embryos. Similar conclusions that normal and abnormal embryos have different morphological parameters were made in the study by Basile et al. [28]. Chavez et al. [29] in their study found that euploid embryos have shorter and more accurate parameters up to the four-cell stage embryo.

For this study, we chose to analyze the morphokinetic parameters cc2, s2, t5 and tSB and the occurrence of multinucleations in two-cell embryos (MN2) and four-cell embryos (MN4). These morphokinetic factors and the occurrence of multinucleations were selected based on our previous unpublished research, which determined that these parameters may be related to embryonic ploidy, and the results of other authors’ work on a similar topic. Efforts have been made to cover these markers throughout the culture of embryos from 2bb to the blastocyst stage. Our selected morphokinetic parameters and morphological markers were also evaluated in the following studies: t5 [28], t5, cc2 [7], s2, [7,30] cc2, s2 [29,30], tSB [30,31,32], and were significantly shorter in euploid embryos than in aneuploid embryos.

The researchers in the study led by Del Carmen Nogales [33] found that morphokinetic parameters may depend on the type of chromosomal abnormality, i.e., embryos with a high degree of chromosomal abnormalities behave differently than euploid embryos. In contrast, embryos with trisomy behave very similarly to euploid embryos. Other studies have found that parameters in implanted/non-implanted embryos [34,35] and euploid/aneuploid embryos [36] do not differ.

Some studies has focused on the assessment of morphokinetic parameters in embryos with varying degrees of mosaicism, because with increasing accuracy of PGT-A (NGS, hr-NGS), the number of detected mosaicist embryos increases. When comparing the groups of euploid embryos and low-level mosaic embryos, no significant differences in morphokinetic parameters were found. In the group of high-level mosaic embryos, some morphokinetic parameters were delayed (e.g., t5, t8, cc3) and a higher incidence of multinucleations was found [37]. Another study shows that morphokinetic parameters of embryonic mosaics overlap the timing of cell division of euploid and aneuploid embryos [38].

Some studies have found that morphokinetic parameters may be influenced by non-genetic factors: for example, the quality of culture media [39], stimulation protocols [40], the fertilization method [41], obesity [42], smoking [43] or oxygen concentration [44].

From the group of PGT-A normal embryos that led to clinical pregnancy, time intervals were determined that should predict embryos with high implantation potential: cc2 = 11.27 h to 11.70 h, 48.10 hpi to 49.52 hpi, tSB = 92.88 hpi to 94.58 hpi.

Similar intervals for these parameters have been published for embryos with the potential to reach the blastocyst stage by Wong et al. [2], when cc2 was set at 7.8–14.3 h, Meseguer et al. [7], who determined that cc2 ≤ 11.9 h, or Cruz et al. [8], who determined a time interval of 48.8–56.6 h for parameter t5. Campbell et al. [31] found an interval for the parameter tSB <96.2 h for embryos with a low risk of aneuploidy.

Multinucleations present in early embryo blastomeres are a common embryonic abnormality [43,45]. Aneuploidy or mosaicism occurred in 50–100% of genetically examined embryos with multinucleations [46]. However, it is likely that embryos, based on a self-correction mechanism, are able to repair multinucleated blastomeres, and after division, two daughter euploid blastomeres with a single nucleus are formed. It is clear that most embryos with multinucleations in blastomeres are aneuploid or mosaicist; however, due to the “self-correction” mechanism, some embryos may eventually be euploid [47]. However, it is clear that the incidence of two-cell stage multinucleations reduces the rate of embryo implantation after ET as well as the number of clinical pregnancies achieved [11], and increases the number of aneuploidies detected [48]. Interestingly, a higher rate of multinucleations was noted in the two-cell stage embryo and significantly decreased by the transition to the four-cell stage embryo [49]. Our study shows that a more significant prediction of aneuploidy in the embryo is the occurrence of multinucleations in the four-cell stage.

Some morphokinetic parameters and morphological markers may be related to the genetic equipment of the embryo and could be used to select the best embryo, especially in patients who are not indicated for PGT or who for any legal, social or economic reasons do not wish to or cannot have PGT performed.

Morphokinetic parameters and morphological features can simplify laboratory procedures, especially when discarding embryos with abnormal development. However, embryological laboratories do not always use the same validated procedures, instruments, materials and media. For instance, embryonic culture at unreduced oxygen tension (20%) is associated with increased reactive oxygen species (ROS) side effects [50], subsequently with interrupted or delayed embryonic development [51] and insufficient Inner Cell Mass development [52]. The advantage of culturing in a time-lapse system is a strictly monitored temperature, as any temperature fluctuations (exposure to room temperature) can negatively affect the development of the embryo (for example, medium change on the 3rd day of cultivation, so culturing in a one-step medium is more appropriate). If the temperature declines, the cleavage rate will also decrease. Similarly, if the pH value is shifted to the alkaline direction, embryo development may be slowed down [53]. In the case that the culture medium is supplemented with serum, it may affect the appearance of the embryo (vesicle formation, darker appearance) and cause earlier cavitation [54].

Thus, there is a need for similar procedures to be followed during oocyte fertilization and embryo culture to ensure comparable culture conditions. Then, time-lapse results from different workplaces could be better integrated into clinical practice.

## 5. Conclusions

Significant differences were found in the morphological parameters cc2, t5 and tSB and the occurrence of multinucleations (MN) in the stage of two-cell and four-cell embryos between the group of genetically normal embryos and abnormal embryos, and between the group of genetically normal embryos that led to clinical pregnancy after ET and the group of abnormal embryos. From the morphokinetic data found in the PGT-A group of euploid embryos leading to clinical pregnancy, time intervals were determined based on statistical analysis, which should predict embryos with high implantation potential: cc2: 11.3–11.7 h, t5: 48.1–49.5 hpi, tSB: 92.9–94.6 hpi. The incidence of MN in the group of euploid embryos that led to clinical pregnancy in the four-cell embryo stage is 7× lower than in the group of aneuploid embryos. In addition, significant differences were found in the morphokinetic parameters t5 and tSB in the group of euploid embryos leading to clinical pregnancy and in the group of euploid embryos that were not implanted after embryo transfer; based on these results, it is recommended that the parameters t5 and tSB be used as essential parameters in the selection of embryos. In the second phase of embryo selection, it is appropriate to use the morphokinetic parameter cc2 and the occurrence of MN during the first two cell divisions. Based on the proposed use of these morphokinetic parameters and the occurrence of MN, the best euploid embryo for ET can be selected with high probability.

## Figures and Tables

**Figure 1 jcm-10-05173-f001:**
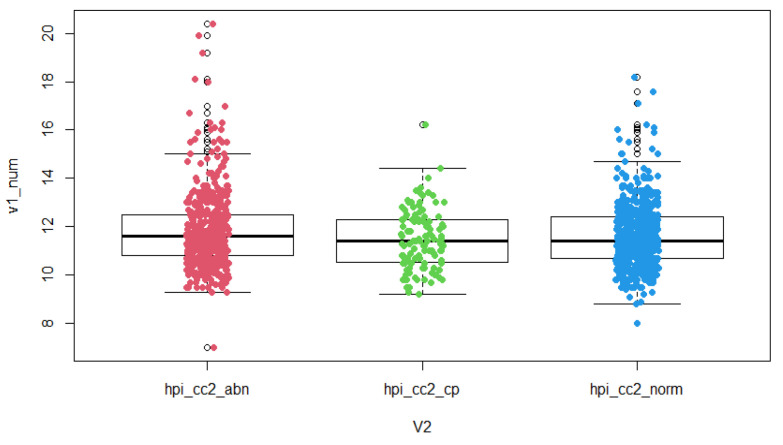
Time-lapse distribution—cc2 in groups of PGT-A abnormal embryos, normal embryos with proven clinical pregnancy and normal embryos. Boxplot is categorized by groups: abn—PGT-A abnormal embryos, cp—PGT-A normal embryos with clinical pregnancy, norm—PGT-A normal embryos, hpi = hours post insemination, v1_num = numerically expressed time, v2 = identification of grouped variables containing the type of embryo.

**Figure 2 jcm-10-05173-f002:**
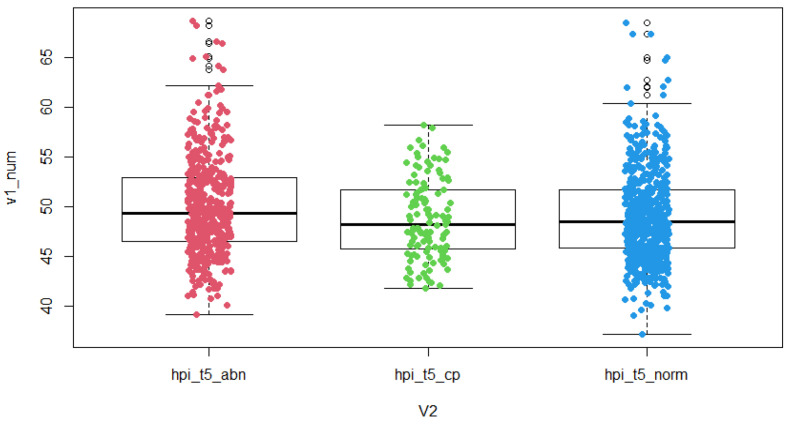
Time-lapse distribution—t5 in groups of PGT-A abnormal embryos, normal embryos with proven clinical pregnancy and normal embryos. Boxplot is categorized by groups: abn—PGT-A abnormal embryos, cp—PGT-A normal embryos with clinical pregnancy, norm—PGT-A normal embryos, hpi = hours post insemination, v1_num = numerically expressed time, v2 = identification of grouped variables containing the type of embryo.

**Figure 3 jcm-10-05173-f003:**
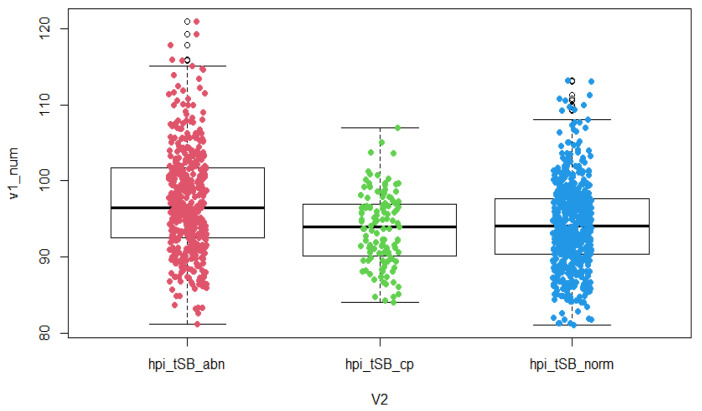
Time-lapse distribution—tSB in groups of PGT-A abnormal embryos, normal embryos with proven clinical pregnancy and normal embryos. Boxplot is categorized by groups: abn—PGT-A abnormal embryos, cp—PGT-A normal embryos with clinical pregnancy, norm—PGT-A normal embryos, hpi = hours post insemination, v1_num = numerically expressed time, v2 = identification of grouped variables containing the type of embryo.

**Figure 4 jcm-10-05173-f004:**
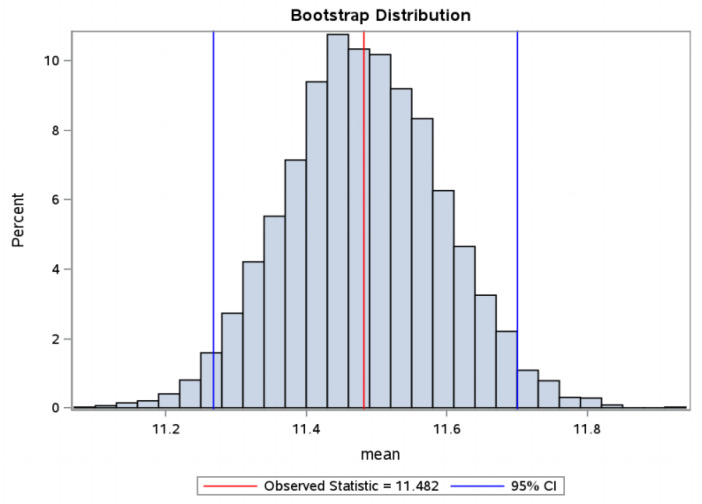
cc2: bootstrap distribution—distribution according to the duration of the 2nd cell cycle in a group of embryos with FHB (in hours). Bootstrap distribution cc2: 95% Lower CI 11.2681, 95% Upper CI 11.7004, FHB = fetal heartbeat.

**Figure 5 jcm-10-05173-f005:**
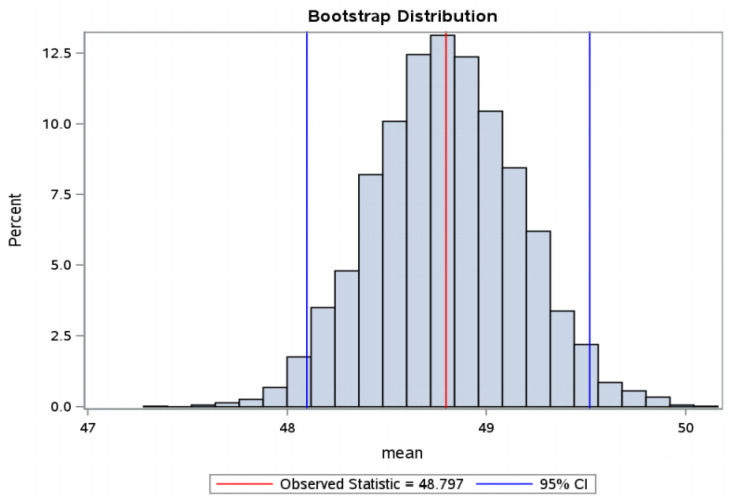
t5—bootstrap distribution—distribution according to the time of reaching the 5-cell embryo stage in the group of embryos with FHB (in hpi—hours post insemination). Bootstrap distribution t5: 95% Lower CI 48.0992, 95% Upper CI 49.5181, FHB = fetal heartbeat.

**Figure 6 jcm-10-05173-f006:**
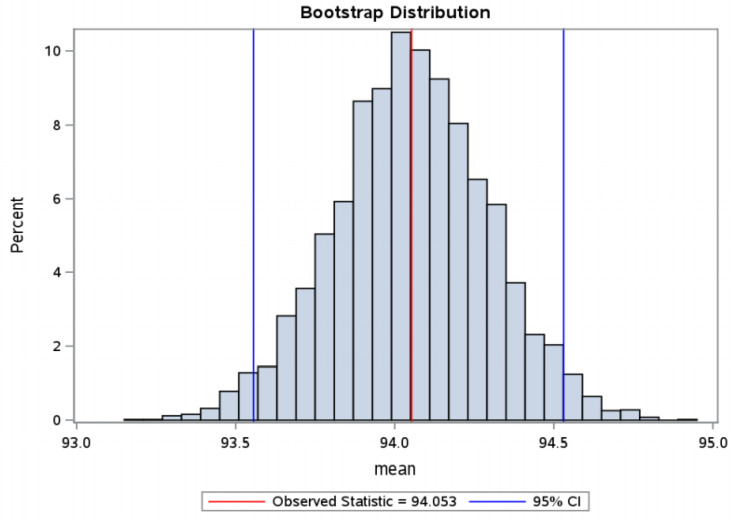
tSB—bootstrap distribution—distribution according to the time of the start of blastulation in the group of embryos with FHB (in hpi—hours post insemination). Bootstrap distribution tSB: 95% Lower CI 92.8790, 95% Upper CI 94.5798, FHB = fetal heartbeat.

**Table 1 jcm-10-05173-t001:** Comparison of morphokinetic parameters in PGT-A groups of normal and abnormal embryos.

Parameter	Normal Embryos	Abnormal Embryos	*p*-Value
	*n* = 552, Mean ± SD	*n* = 475, Mean ± SD	
cc2/h	11.605 ± 1.321	11.856 ± 1.601	*p* = 0.007
s2/h	0.672 ± 0.714	0.841 ± 1.053	*p* = 0.065
t5/hpi	49.151 ± 4.619	50.066 ± 4.931	*p* = 0.002
tSB/hpi	94.051 ± 5.719	97.238 ± 6.737	*p* < 0.0001

h = hours, hpi = hours post insemination, cc2 = duration of the second cell cycle, s2 = time between second and third mitosis, t5 = time to reach 5-cell embryo, tSB = time to initiate blastulation, for parameter s2 median and IQR 0.5 (0.3; 0.8).

**Table 2 jcm-10-05173-t002:** Comparison of morphokinetic parameters in groups of PGT-A normal embryos with achieved clinical pregnancy and abnormal embryos.

Parameter	Normal Embryos with FHB	Abnormal Embryos	*p*-Value
	*n* = 119, Mean ± SD	*n* = 475, Mean ± SD	
cc2/h	11.483 ± 1.222	11.856 ± 1.601	*p* = 0.006
s2/h	0.635 ± 0.746	0.841 ±1.052	*p* = 0.021 *
t5/hpi	48.800 ± 4.004	50.066 ± 4.931	*p* = 0.004
tSB/hpi	93.726 ± 4.789	97.238 ± 6.737	*p* < 0.0001

h = hours, hpi = hours post insemination, cc2 = duration of the second cell cycle, s2 = time between second and third mitosis, t5 = time to reach 5-cell embryo, tSB = time to initiate blastulation, FHB = fetal heartbeat, for parameter s2 median and IQR 0.5 (0.3; 0.8), * Mann–Whitney *U*-test.

**Table 3 jcm-10-05173-t003:** Occurrence of multinucleations in 2-cell stage and 4-cell stage embryos and groups normal embryos (euploid) with proven clinical pregnancy, normal embryos (euploid) and abnormal embryos (aneuploid).

	Normal Embryos with FHB	Normal Embryos
	Odds Ratio 95%CI	Odds Ratio 95%CI
	*p*-Value	*p*-Value
	*n* = 119	*n* = 552
NM2 abnormal embryos (*n* = 475) vs. group	2.013 (1.343; 3.028)	2.184 (1.7; 2.810)
	*p* < 0.0001	*p* < 0.0001
NM4 abnormal embryos (*n* = 475) vs. group	7.061 (4; 13.501)	6.269 (4.609; 8.622)
	*p* < 0.0001	*p* < 0.0001

NM2 = multinucleations in 2-cell stage embryos, MN4 = multinucleations in 4-cell stage embryos, FHB = fetal heartbeat.

## Data Availability

The authors confirm that the data supporting the findings are available from the corresponding author upon reasonable request.

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
