# Peer review of "Differences in Morphokinetic Parameters and Incidence of Multinucleations in Human Embryos of Genetically Normal, Abnormal and Euploid Embryos Leading to Clinical Pregnancy"

_jcm, 2021, doi:10.3390/jcm10215173_

Round 1
Reviewer 1 Report
- Conclusions should be detailed from the text.
- The method used should be supported by references, unless it is innovative.
- Line 69-73: this sentence should be corrected because it is not clearly understood.
Author Response
Thank you for your comments.
- Conclusions have been added to the text
We have added a comparison of groups of euploid embryos with proven clinical pregnancy (n=119) and a group of embryos without pregnancy after ET (n=99). There were significantly differences in the morphokinetic parameter t5 and tSB (p = 0.037 t5, p = 0.036 tSB).
Based on this result, we can recommend to prioritize parameters t5 and tSB in embryo selection (added to the results and conclusions).
- References were added to the methods used
- The text on lines 69-73 has been modified:
With the implementation of trophectoderm blastocyst biopsy on the fifth or sixth day of the culture (not only one or two blastomers on the third day of embryo development) in IVF treatment cycles with PGT-A and using whole embryo genome amplification, a higher pregnancy rate was observed; especially in cycles with vitrified biopsied embryo transfer compared to cycles with fresh embryo transfer in the morning of the sixth day of the culture (with a biopsy in the morning of the fifth day of the culture) [18].

Reviewer 2 Report
This retrospective study collected the data of 1027 morphologically high-quality embryos from 296 patients. Array-based comparative genomic hybridization (aCGH) and next generation sequencing (NGS) platforms were applied to determine individual blastocysts with a normal or an abnormal ploidy status. The correlations of morphokinetic parameters (cc2, s2, t5, and tSB) and the occurrence of multinucleations (MN2 and MN4) with embryo ploidy were evaluated using time-lapse monitoring. The results showed that cc2, t5, tSB, and the multinucleation appearances were significantly different between genetically normal and abnormal blastocysts. The authors suggested these parameters could be the applied for selection of embryo with high pregnancy potential. Several questions in the following paragraph need further consideration.
- The time-lapse monitoring could offer various parameters about developmental kinetics, dysmorphisms, and embryo morphology. It is not clear why authors decide to use cc2, s2, t5, tSB, and the occurrence of multinucleations for analysis. Even though the cc2, t5, tSB and MN occurrences might differ in the euploid and aneuploid embryos, the authors may need to suggest how to integrate these factors for embryo selection.
- aCGH and NGS platforms were used in this study. It is unknown about the exact definition for genetically normal and abnormal embryos. How did authors assign the mosaic embryos to the normal or the abnormal group?
- Several studies (such as Martín et al., 2021; Lee at al., 2019) have demonstrated the correlation between embryo ploidy status and time-lapse parameters. Please update and include these studies for comparison and discussion.
- Logistic regression was used to revealed the OR for MN occurrences between ploidy groups. In retrospective data collection, it may need to take the confounding factors in to account. Besides, the statistical analysis need to consider that multiple embryos (repeated measures) may be derived from an individual patient.
- In M&M, please have explanation for KET and the methods for preparation of HRT and natural cycles.
Author Response
Thank you for the comments and suggestions.
- Morphokinetic factors (cc2, s2, t5, tSB) and the occurrence of multinucleations were selected based on our previous unpublished research, which determined that these parameters could be related to embryonic ploidy and the results of works by other authors on a similar topic (Basile et al. , 2014, Meseguer et al, 2011, Minasi et al., 2016, Chavez et al, 2012, Campbell et al, 2013, Mumusoglu, et al., 2017, Munne et al, 2006, Yilmaz et al, 2014, Ergin et al. al, 2014, Ambroggio et al, 2011, Balakier et al, 2016). Efforts have been made to cover these markers throughout the culture of embryos from 2bb to the blastocyst stage.
Due to the fact that we also (after sending the article) compared groups of euploid embryos with proven clinical pregnancy and a group of embryos without pregnancy after ET. There were significantly differences in the morphokinetic parameter t5 and tSB (p = 0.037 t5, p = 0.036 tSB).
Based on this result, we can recommend to prioritize parameters t5 and tSB in embryo selection, then to prefer embryos without multinucleation on 4-cell stage (MN4 - which occur 7 times more often in aneuploid embryos) and in the third round use cc2 or MN2 as the weakest parameter.
We have summarized these results in the chapter "conclusions"
- In our study, a normal embryo is defined up to a maximum of 20% mosaicism according to PGDIS recommendations (May 27, 2019),
added to the introduction and discussion
- The recommended studies about mosaicism were included in the Discussion
-
We have been discussed this topic previously in the consortium and agree that in clinical studies is common to use confound variablles. Due to the fact that a homogeneous group of subjects was selected (age, no other diseases ...) using in and out conditions, confusing variables were not used in our situation. Moreover to the comment regarding hierarchical models. We have treated each embryo as a separate subject regardless of the patient. Each embryo has its own probability that is affected by the carrier but in concern of logistic regression doesn't affect OR. The repeated measures were not used. We will be pleased to recalculate when the reviewer will request to do it so and will not agree with our conclusions.
-
KET is "kryoembryotransfer" = frozen embryo transfer (We will change KET to FET), preparation of HRT and natural cycle was explanated in the M&M

Reviewer 3 Report
In the manuscript, the Authors tried to find morphokinetic parameters, which could help to choose the embryo with the highest chance to result in pregnancy after embryotransfer. Especially, they focused on the prediction of genetic aberrations and the possibility of replacing preimplantation genetic testing by noninvasive methods. As more and more couples need in vitro fertilization and there are high expectations that those procedures will result in the birth of a healthy child, such research is of high importance.
The manuscript is well written and properly structured. The aim of the study is clear, although it is not explained why this research was performed while similar studies have been already published.
Although the Authors stated that they identified morphokinetic parameters according to which the best euploid embryo for ET can be selected with high probability, their clinical usefulness is doubtful. Comparison of the thresholds proposed based on 95% CI in embryos leading to pregnancy (Fig. 4, 5, 6) with values obtained in other groups (Figs. 1, 2, 3) clearly shows that many abnormal embryos would present the values established here as predictive for ‘the best’ embryos. Therefore, many false decisions can be made. Inversely, many embryos that led to the pregnancy would be discarded based on the proposed thresholds.
Moreover, as Authors stated in the discussion, similar study have been performed previously, leading to different results and conclusions. As embryo morphokinetic may be influenced by many other factors, e.g., culture media, the results are not universal and therefore their utility is limited. This was not elaborated sufficiently in the discussion and must be highlighted better than only in Line 286.
Also, in my opinion, the allocation of embryos into three experimental groups is not completely right. At the beginning, embryos were divided into two groups based on PGT: aneuploid and euploid. Then a third group was subtracted from the second one: euploid embryos resulted in pregnancy. The results of comparison of I and II and I and III groups are almost the same – is it because group III includes in II? I think it would be better for statistical analysis to create three groups which are not overlapping: I – aneuploid, II – euploid not resulting in pregnancy, III – euploid resulting in pregnancy. It would be interesting to check if euploid embryos not resulting in pregnancy can be differentiated from those leading to pregnancy.
There are other minor remarks:
Line 60-61: ‘normal culture’ à better ‘traditional assesment’
Line 64: repetition ‘is’
Line 71: something is missing
Line 77: the use of abbreviations for the first time without explanation
Lines 106-107: details about quality grading in brackets?
Line 133: the use of abbreviations for the first time without explanation
Line 160 and further: instead of normal – euploid, the use of euploid/aneuploid should be enough
Table 1 – the units (‘h’, ‘hpi’) better in square brackets
Figures 1 and 2 – what is ‘v1_num’?
To sum up: the manuscript provides some new data on the important topic, however, it needs to be revised.
Author Response
Thank you for your comments and suggestions.
The main goal of this observational study is to identify potential morphokinetic parameters for the detection best euploid embryo Our ultimate goal is not to determine a standard threshold for these parameters for such a thing we do not keep enough observations. The robust 95% CI has been created to study the distribution of those selected parameters for further research.
We broadened the discussion on cultivation conditions (line 286...)
We have added a comparison of groups of euploid embryos with proven clinical pregnancy (n=119) and a group of embryos without pregnancy after ET (n=99). There were significantly differences in the morphokinetic parameter t5 and tSB (p = 0.037 t5, p = 0.036 tSB).
Based on this result, we can recommend to prioritize parameters t5 and tSB in embryo selection (added to the results and conclusions).
We have corrected minor errors in the text
We will delete the graph description "v1_num"

Round 2
Reviewer 2 Report
There are no further questions.